# SARS-CoV-2 and Coronavirus Disease 2019: What We Know So Far

**DOI:** 10.3390/pathogens9030231

**Published:** 2020-03-20

**Authors:** Firas A. Rabi, Mazhar S. Al Zoubi, Ghena A. Kasasbeh, Dunia M. Salameh, Amjad D. Al-Nasser

**Affiliations:** 1Department of Clinical Sciences, Faculty of Medicine, Yarmouk University, Irbid 21163, Jordan; 2Department of Basic Medical Sciences, Faculty of Medicine, Yarmouk University, Irbid 21163, Jordan; mszoubi@yu.edu.jo; 3School of Medicine, Yarmouk University, Irbid 21163, Jordan; 2014999093@ses.yu.edu.jo (G.A.K.); 2014799006@ses.yu.edu.jo (D.M.S.); 4Department of Statistics, Faculty of Sciences, Yarmouk University, Irbid 21163, Jordan; amjadn@yu.edu.jo

**Keywords:** Coronavirus, COVID-19, SARS, SARS-CoV-2, Wuhan, pandemic

## Abstract

In December 2019, a cluster of fatal pneumonia cases presented in Wuhan, China. They were caused by a previously unknown coronavirus. All patients had been associated with the Wuhan Wholefood market, where seafood and live animals are sold. The virus spread rapidly and public health authorities in China initiated a containment effort. However, by that time, travelers had carried the virus to many countries, sparking memories of the previous coronavirus epidemics, severe acute respiratory syndrome (SARS) and Middle East respiratory syndrome (MERS), and causing widespread media attention and panic. Based on clinical criteria and available serological and molecular information, the new disease was called coronavirus disease of 2019 (COVID-19), and the novel coronavirus was called SARS Coronavirus-2 (SARS-CoV-2), emphasizing its close relationship to the 2002 SARS virus (SARS-CoV). The scientific community raced to uncover the origin of the virus, understand the pathogenesis of the disease, develop treatment options, define the risk factors, and work on vaccine development. Here we present a summary of current knowledge regarding the novel coronavirus and the disease it causes.

## 1. Introduction

Coronaviruses, named for the crown-like spikes on their surface (Latin: corona = crown), are positive-sense RNA viruses that belong to the Coronvirinae subfamily, in the Coronaviridae family of the Nidovirales order [1]. They have four main subgroups—alpha, beta, gamma, and delta—based on their genomic structure. Alpha- and betacoronaviruses infect only mammals, usually causing respiratory symptoms in humans and gastroenteritis in other animals [2,3]. Until December of 2019, only six different coronaviruses were known to infect humans. Four of these (HCoV-NL63, HCoV-229E, HCoV-OC43 and HKU1) usually caused mild common cold-type symptoms in immunocompetent people and the other two have caused pandemics in the past two decades. In 2002–2003, the severe acute respiratory syndrome coronavirus (SARS-CoV) caused a SARS epidemic that resulted in a 10% mortality. Similarly, the Middle East respiratory syndrome coronavirus (MERS-CoV) caused a devastating pandemic in 2012 with a 37% mortality rate.

In late 2019, a cluster of pneumonia cases in Wuhan City, Hubei Province, China were identified as with a novel betacoronavirus, first called the 2019 novel coronavirus (2019-nCov) and often referred to as the Wuhan coronavirus. When the genomics of the 2019-nCov was sequenced, it shared 79.5% of the genetic sequence of the SARS-CoV that caused the 2002–2003 pandemic [4] and the International Committee on Taxonomy of Viruses renamed the 2019-nCov as SARS-CoV-2 [5]. Patients began to present in November and December with various degrees of respiratory distress of unknown etiology and treated at the time as possible influenza infections. As it became apparent that most cases had a shared history of exposure to the Huanan Seafood Wholesale Market (the so-called “wet market”), the Wuhan local health authority issued an epidemiologic alert on 30 December 2019 and the wet market was closed. About a week later, on 9 January 2020, Chinese researchers shared the full genetic sequence of the novel coronavirus, now called SARS-CoV-2 [6]. Since the novel coronavirus was recognized, the disease it caused was termed coronavirus disease 2019 (CoVID-19), and several reports on the clinical presentation, epidemiology, and treatment strategies have been published [7,8,9,10]. In addition, several websites have been setup to track the epidemic and the case detection rate, which are being updated as often as hourly [11,12,13,14]. On 30 January 2020, the World Health Organization (WHO) declared the COVID-19 outbreak to be a global public health emergency, sixth after H1N1 (2009), polio (2014), Ebola in West Africa (2014), Zika (2016) and Ebola in the Democratic Republic of Congo (2019), and on 11 March 2020, the WHO characterized COVID-19 as a pandemic [15]. The timeline of events is summarized in Figure 1.

## 2. The Origin of SARS-CoV-2

All coronaviruses that have caused diseases to humans have had animal origins—generally either in bats or rodents [16]. Previous outbreaks of betacoronaviruses in humans involved direct exposure to animals other than bats. In the case of SARS-CoV and MERS-CoV, they were transmitted directly to humans from civet cats and dromedary camels respectively (Figure 2).

The SARS-related coronaviruses are covered by spike proteins that contain a variable receptor-binding domain (RBD). This RBD binds to angiotensin-converting enzyme-2 (ACE-2) receptor found in the heart, lungs, kidneys, and gastrointestinal tract [18] thus facilitating viral entry into target cells. Based on genomic sequencing, the RBD of SARS-CoV-2 appears to be a mutated version of its most closely related virus, RaTG13, sampled from bats (*Rhinolophus affinis*) [19]. It is, therefore, believed that the SARS-CoV-2 also originated from bats and, after mutating, was able to infect other animals. The mutation increased the RBD affinity to ACE-2 in humans, but also other animals such as ferrets and Malayan pangolins (*Manis javanica*; a long-snouted, ant-eating mammal sold illegally for use in traditional Chinese medicine), but also decreased the RBD affinity to ACE-2 found in rodents and civets. The pangolin is believed to be the intermediate host of SARS-CoV-2 [17].

There was some early speculation that SARS-CoV-2 emerged from a manmade manipulation of an existing coronavirus, but there is no evidence to support such a theory. In fact, Anderson et al. suggest that the particular mutation that was found in the RBD of SARS-CoV-2 is different to what would have been predicted based on previously used genetic systems. The authors, however, stated that “it is currently impossible to prove or disprove the other theories of [the SARS-CoV-2] origin [19]”. 

## 3. Pathogenesis and Clinical Presentation

Since SARS-CoV and SARS-CoV-2 are so similar, the biochemical interactions and the pathogenesis are likely similar. Binding of the SARS-CoV to the angiotensin-converting enzyme 2 (ACE-2) receptors in the type II pneumocytes in the lungs triggers a cascade of inflammation in the lower respiratory tract [20]. It has been demonstrated that when the SARS spike protein binds to the ACE-2 receptor (Figure 3A), the complex is proteolytically processed by type 2 transmembrane protease TMPRSS2 leading to cleavage of ACE-2 and activation of the spike protein (Figure 3B) [21,22], similar to the mechanism employed by influenza and human metapneumovirus, thus facilitating viral entry into the target cell (Figure 3C). It has been suggested that cells in which ACE-2 and TMPRSS2 are simultaneously present are most susceptible to entry by SARS-CoV [23]. Early indications are that SARS-CoV-2 virus also requires ACE-2 and TMPRSS2 to enter cells [24].

Viral entry and cell infection trigger the host’s immune response, and the inflammatory cascade is initiated by antigen-presenting cells (APC). The process starts with the APC performing two functions: (1) presenting the foreign antigen to CD_4_^+^-T-helper (Th1) cells, and (2) releasing interleukin-12 to further stimulate the Th1 cell. The Th1 cells stimulate CD_8_^+^-T-killer (Tk) cells that will target any cells containing the foreign antigen. In addition, activated Th1 cells stimulate B-cells to produce antigen-specific antibodies.

In the first published review of the clinical presentation of 41 patients admitted to hospital with COVID-19 [9], 98% of patients had a fever, 76% had a cough, and 55% had shortness of breath on admission. However, those admitted may have had less severe symptoms for 2 to 14 days prior to presentation, during which they were likely contagious. By the time patients developed shortness of breath, they had been sick for an average of eight days. Once admitted to the hospital, all patients developed clinical pneumonia supported by chest CT findings, and 13 of the 41 patients (32%) developed hypoxic respiratory failure necessitating ICU admission. Four patients (10%) required mechanical ventilation, two of which received extracorporeal membrane oxygenation due to refractory hypoxia. In total, six patients died, giving a case fatality rate (CFR) of 15% and triggering panic that quickly spread worldwide. While early media reports suggested that deaths were more likely in patients with comorbid conditions, of the 41 patients described in the Chinese review, only 38% had comorbid conditions and the average age was 49.

## 4. Epidemiology

As of March 16, 0700 GMT, there were 169,930 confirmed cases, about half of which (80,860 cases, 47.6%) were within mainland China. About 18% of ill people had severe disease, and 82.0% had mild disease and a total of 889 tested-positive cases were asymptomatic [13,25]. While initially confined to China among those who visited the Wuhan wet market, over the course of about 3 months the SARS-CoV-2 has to date been confirmed in 157 countries and one cruise ship [13]. The Chinese CDC published the epidemiologic characteristics of the COVID-19 outbreak as of 11 February 2020 (Table 1) [26]. Initial data suggests that the majority of patients (73%) were over age 40 years, and that the risk of death increases with age. No deaths were reported in patients younger than 10 years old, and only 2.6% of the total fatalities were in patients younger than 40 years of age.

After mainland China, the next area with the highest number of confirmed cases is Italy, where as of 16 March 2020, there were 24,747 reported cases, and Iran where 13,938 cases have been confirmed. To date, there have been 6522 deaths worldwide (3.83%), 3213 of which (49.2%) were within mainland China [11,13]. Due to aggressive containment strategies in China, including a mass quarantine of the entire 11 million population of Wuhan, the acceleration of new cases in China has slowed whereas that outside of China has increased. As of March 2nd, the number of daily new cases outside of China was nine times higher than those within China. Many countries have instituted travel bans and/or quarantine procedures for incoming travelers. Closures of public schools and social gatherings have been instituted in many countries in an effort to contain the spread of COVID-19 and decrease the public health burden [27,28] and the CDC has released recommendations on school closure criteria [29].

In comparison, the 2002 SARS pandemic, which also originated in China, resulted in 8096 people infected and 774 deaths (9.6%). On the other hand, the 2012 MERS pandemic infected 2494 people causing 858 deaths (34.4%). Therefore, although MERS and SARS had higher mortality, the much larger number of people infected with SARS-CoV-2, and the rate at which the number is increasing, raises red epidemiologic flags.

To assess the magnitude of the risk posed by the SARS-CoV-2, we review four parameters that we believe important: the transmission rate, the incubation period, the case fatality rate (CFR), and the determination of whether asymptomatic transmission can occur.

### 4.1. Transmission Rate

The reproduction number, or “R naught” (R0), is a mathematical term that defines contagiousness [30]. Specifically, it is the number of people that one sick host can infect. If the R0 is less than one the disease will disappear. If the R0 ≥ 1 then the disease will spread between people. Estimates of the R0 of SARS-CoV-2 have ranged from 2.24 to as high as 3.58 [31] although the World Health Organization estimates it is between 1.4 and 2.5 [32]. For the purposes of comparison, the mean R0 for seasonal influenza is between 1.1 and 2.3 (variable by region and immunization rates), whereas for SARS was between 1 and 2.75. The slightly higher R0 for SARS-CoV-2 may be because it has a longer prodromal period, increasing the period during which the infected host is contagious. 

Coronaviruses are generally thought to be spread most often by respiratory droplets, not to be confused with airborne transmission [33]. Droplets are larger and tend to fall to the ground close to the infected host and only infect others if the droplet is intercepted by a susceptible host prior to landing. Droplet transmission is typically limited to short distances, generally less than 2 m. However, the airborne route involves much smaller droplets that can float and move longer distances with air currents. Under certain humidity and temperature environments, airborne droplets can remain in flight for hours. Generally, pathogens that are transmissible via the airborne route have higher R0, because infected particles can remain in the air long after the infected individual has left the premises. This airborne route occurs, for example, in measles (R0 between 12 and 18 [34]) and chicken pox (R0s between 3.7 and 5.0 [35]). 

Once infected droplets have landed on surfaces, their survivability on those surfaces determines if contact transmission is possible. Based on our current understanding from other betacoronaviruses, including SARS and MERS, coronaviruses can survive, and remain infectious, from 2 h up to 9 days on inanimate surfaces such as metal, glass, or plastic, with increased survival in colder and dryer environments [36,37,38]. For this reason, the Chinese government has been reported to be disinfecting and even destroying cash in an effort to contain the virus [39]. Reassuringly, cleansing of surfaces with common biocidals such as ethanol and sodium hypochlorite is very effective at inactivation of the coronaviruses within 1 min of exposure [36]. 

The timing of maximum infectivity is currently being assessed. A small study of 17 patients showed that nasal viral load peaks within days of symptom onset, suggesting that transmission of disease is more likely to occur early in the course of infection [40].

### 4.2. Incubation Period

Understanding incubation periods is very important as it allows health authorities to introduce more effective quarantine systems for suspected cases. The best current estimates of the SARS-CoV-2 infection range from 2 to 14 days. Analysis of the first 425 cases of COVID-19 in Wuhan a mean incubation period of 5.2 days [41]. A later report, based on 1324 cases, reported a mean incubation period of 3.0 days [42]. Yet another report, on 88 cases who traveled to Wuhan between 20 and 28 January, had incubation period ranges from 2.1 to 11.1 days, with a mean of 6.4 days [43].

### 4.3. Case Fatality Rate

To calculate the case fatality rate (CFR) of an infection, one must divide the mortality number (M) by all those who were infected. The total number of those infected includes those who were infected and recovered without presentation (I_r_), infected and presented to a health care facility (I_p_), and infected and died (I_d_). The CFR would be M/(I_r_ + I_p_ + I_d_). Clearly, one must have an accurate estimation of each of these parameters to accurately determine the CFR of COVID-19. While the (M) is generally easier to count, and a focus of media, the denominator can take much longer to calculate. During the early phases of a deadly epidemic, the number of those who were infected and recovered (I_r_) is not yet known, since only those who were infected and became seriously ill are recognized and tested. In addition, because this is a novel virus, there were no existing detection methods, so early deaths due to clinical entities such as influenza, for example, may have been mis-attributed to CoVID-19. The viral genome was published about 2 weeks after the start of the outbreak, and PCR analysis was quickly used to diagnose suspected cases [6]. Public health officials can now test suspected cases, especially close contacts of known cases, and others with mild symptoms, but the testing capabilities can become saturated, potentially limiting the ability to get an accurate estimation of I_p_. For example, the initial ability of the Wuhan health authority was limited to 200 tests per day, but that number has grown to 4196 tests per day [44]. The combination of these factors leads to a gross underestimation of the denominator of the CFR calculation, and thus an exaggeration of the mortality. Until we are able to accurately represent I_r_ and I_p_, it is currently impossible to precisely estimate the CFR of SARS-CoV-2. 

However, during the course of a potentially fatal pandemic, an accurate estimation of CFR is important. While it is tempting to estimate the CFR by dividing the number of known deaths by the total number of confirmed cases, the resulting number may be off by orders of magnitude, especially since infected individuals at one point in time may die *x* days later. Using the lag period approach and dividing the current number of deaths to the number of cases x days ago may be a more acute estimator of CFR. Nucleuswealth.com applied this method by using the number of deaths at any particular day and dividing by number of cases 4, 8, or 12 days prior. As seen in Figure 4, as time progresses, whether whichever number of days is used for x, the CFR seems to converge at just under 5% for cases within Hubei, and about 0.8% for cases outside of Hubei [14]. The higher mortality in Wuhan may be overestimated because early in the course of this epidemic, viral testing was limited to only the severe cases. However, the China National Health Commission admits that Wuhan has a relative lack of medical resources, which may have contributed to the higher mortality rate.

### 4.4. Asymptomatic Transmission

Infection transmission by asymptomatic individuals can make control of disease spread challenging. Since late January, SARS-CoV-2 transmission from infected but still asymptomatic individuals has been increasingly reported [45,46]. Assessment of the viral loads in symptomatic individuals not only showed that the viral loads peak within the first few days of symptoms, but also that asymptomatic patients can have a similarly high viral load without showing symptoms [40]. It was suggested that viral testing should no longer be limited to symptomatic individuals, but also include those who have traveled to affected areas [47].

## 5. Risk Factors for Mortality

At such an early phase of the COVID-19 pandemic, it is difficult to accurately describe the populations most at risk, especially when teasing out risk factors for infection from risk factors for death from disease. Early on, it became clear that those who have visited the Wuhan wet market were most at risk of infection, but the population visiting the market is not an accurate reflection of the general population. The Chinese CDC published the epidemiologic characteristics of the COVID-19 outbreak along with associated risk factors for death [26]. 

The largest risk factor for death is age. Other risk factors include male sex and the presence of comorbid conditions (Table 2). However, in addition to real age-specific mortality, the age-based risk could reflect underlying comorbidities among the elderly and the distribution of the underlying population in Wuhan, where the outbreak initiated. 

With what we know about the pathogenesis of the SARS-CoV virus, it seems reasonable to assume that those with higher levels of ACE-2 receptors may be at greatest risk. There was some speculation that the expression of ACE-2 receptors may be linked to race, specifically after an early report suggested that Asian males had higher ACE-2-expressing cell ratios than white and African Americans [48]. However, the sample size contained only eight different individuals (five African Americans, two whites, and one Asian) and extrapolating those findings to a whole race is impractical. Yet, in another study assessing ACE-2 receptor expression in tissues of 224 patients with lung cancer, there were no significant disparities in ACE-2 gene expression between racial groups (Asian vs. Caucasian), age groups (older or younger than 60 years old), or gender groups (male vs. females) [49]. ACE-2 gene expression was, however, significantly elevated in smokers suggesting that smoking history should be considered in identifying susceptible populations. Since smoking in China is predominantly a male attribute (54% of men, 2.6% of women) [50], this may help to explain the gender difference seen in the hospitals in China.

Early in the COVID-19 epidemic, it appeared that children were a protected group, but this may have been because they were less likely to have frequented the Wuhan wet market, or because they were more likely to have asymptomatic or mild disease and thus less likely to have been tested. COVID-19 has affected infants as young as 1 month of age [51], most with mild or asymptomatic disease. There have been no reported cases of adverse infant outcomes for mothers who developed COVID-19 during pregnancy. 

Second to the Hubei population, the other population at increasing risk is healthcare workers. As of February 17, 2020, total of 1716 healthcare workers in China have been infected, five of whom fatally [25].

## 6. Treatment

The current best strategy of treatment of patients with COVID-19 is purely supportive. Clinicians and intensive care specialists are applying much of what they have learned during the SARS epidemic to guide current therapy of COVID-19. Recommendations for admission to critical care units, guidelines for infection control, and procedures to minimize nosocomial transmission are being established [52]. However, there are several fronts that are being studied to develop targeted treatments. 

The most efficient approach to the treatment of COVID-19 is to test whether existing antiviral drugs are effective. In previous betacoronavirus epidemics, several antiviral drugs, such as ribavirin, interferon, lopinavir-ritonavir, and darunavir/cobicistat (prezcobix) were tested, with some showing promising in vitro results [53]. Remdesivir, an adenosine analog used against RNA viruses (including SARS and MERS-CoV), was a candidate Ebola treatment with promising in vitro results but disappointing in vivo effects against Ebola [54,55]. There is currently in vitro evidence that remdesivir may be effective in controlling SARS-CoV-2 infection [56]. In fact, compassionate use of remdesivir was employed in the treatment of the first COVID-19 case in the United States, during a period of rapid clinical deterioration, and within one day there was dramatic improvement of the clinical condition [57]. Randomized double-blinded, placebo-controlled clinical trials are currently underway in China and USA to evaluate the efficacy of remdesivir and initial results are expected by the end of April 2020 [58,59].

Other existing drug candidates include chloroquine and camostat mesylate. Chloroquine is a widely used anti-malarial drug that is known to block virus-cell fusion and has been shown to interfere with the glycosylation of SARS-CoV and ACE-2 cellular receptors, rendering the ACE-2-SARS-CoV interaction less efficient [60]. There is also in vitro evidence that chloroquine may be effective in preventing SARS-CoV-2 cellular entry [56]. Camostat mesylate, also known as FOY 305 [61], was initially developed and currently approved for the treatment of chronic pancreatitis in Japan [62,63]. Camostat mesylate targets the TMPRSS2 protease, theoretically preventing viral entry. Researchers in Germany showed that camostat mesylate reduced the amount of SARS-CoV-2 viral replication [64]. 

A simple but very effective treatment modality is the use of convalescent plasma, or serum from patients who have recovered from the virus, to treat patients. Patients with resolved viral infection will have developed a specific antibody response which may be helpful in neutralizing viruses in newly infected individuals. This modality was successfully employed during the 2014–2015 Ebola outbreak [65,66]. However, the use of convalescent sera is of limited benefit in an outbreak situation since the exponential growth of infected patients exceeds the ability of previous patients to provide donor plasma.

The recent finding that SARS-CoV-2 binds to the same ACE-2 receptors targeted by the 2002 SARS-CoV [24] opens up the possibility of using the previous research on the 2002 SARS epidemic and applying it to COVID-19. The first strategy would be to employ either a small receptor-binding domain (RBD) or a neutralizing antibody targeting the ACE-2 receptor, thus blocking the binding of S protein and preventing virus entry into cells. Initial in vitro results have shown promising results [67,68] and specific monoclonal antibodies are being contemplated as candidates for treatment [69,70]. The main limitation of using RBDs or antibodies is that the treatment must be given within a specific time window, before the initiation of viral replication [20]. In addition, the side effects of ACE-2 blockade, especially since ACE-2 is also present in non-pulmonary tissue, must be understood and minimized before implementation. In addition, finally, the turnover of ACE-2 receptors would influence how often the therapeutic RBD or antibody would have to be administered. 

A second strategy is to create an ACE-2-like molecule that would bind to the S protein of the coronavirus itself. Again, research in to the 2002 SARS virus demonstrated that soluble ACE-2 proteins blocked the SARS virus from infecting cells in vitro [68,71]. The additional benefit to using this strategy lies in the possible prevention of S protein-mediated ACE-2 shedding that has been shown to induce the pulmonary edema characteristic of SARS [72,73]. A phase II clinical trial of recombinant ACE-2 in ARDS reported significant modulation of inflammatory proteins, but no significant differences in respiratory parameters [74]. Further research is necessary to assess if the animal studies will translate to clinical benefit.

There are currently more than 80 clinical trials to test a variety of potential SARS-CoV-2 treatments [75].

## 7. Vaccine Development

The long-term goal of SARS-CoV-2 research is developing an effective vaccine to yield neutralizing antibodies. The National Institutes of Health in the US, and Baylor University in Waco, Texas, are working on a vaccine based on what they know about the coronavirus in general, using information from the SARS outbreak. In addition, the recent mapping of the SARS-CoV-2 spike protein may pave the way for more rapid development of a specific vaccine [76]. Of interest is the use of a relatively new vaccine technology, RNA vaccines that have the ability to elicit potent immune responses against infectious diseases and certain cancers [77,78]. Traditional vaccines stimulate the production of antibodies via challenges with purified proteins from the pathogens, or by using whole cells (live, attenuated vaccines). While very effective, the creation of new vaccines can take years. Alternatively, RNA-based vaccines use mRNA that upon entering cells, are translated to antigenic molecules that in turn, stimulate the immune system. This process has been used effectively against some cancers [79,80], and clinical trials are underway for several other cancers [81]. In addition, the production of RNA-based vaccines is more rapid and less expensive than traditional vaccines, which can be a major advantage in pandemic situations. Clinical trials for an mRNA-based SARS-CoV-2 vaccine are currently underway [82]. Study subjects will receive the mRNA vaccine in two doses, 28 days apart and the safety and immunogenicity will be assessed.

## Figures and Tables

**Figure 1 pathogens-09-00231-f001:**
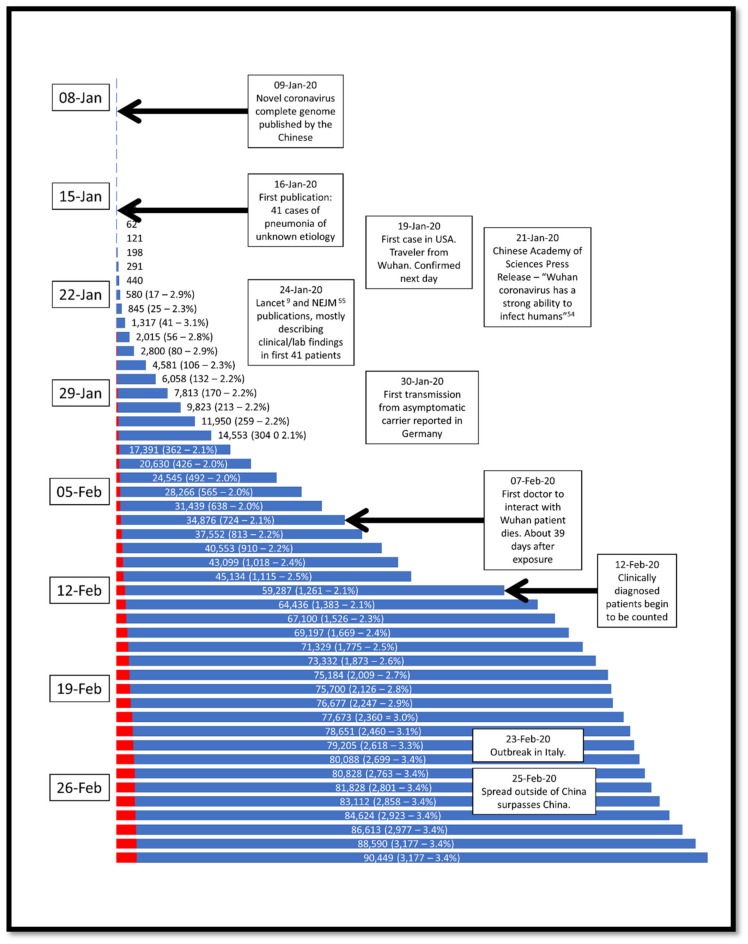
Timeline of the SARS-CoV-2 epidemic, with significant dates noted. Each blue bar represents the cumulative number of COVID-19 patients diagnosed to that day, and the red bar the cumulative number of deaths. At each date, the actual numbers are present. Data from Worldometer [13].

**Figure 2 pathogens-09-00231-f002:**
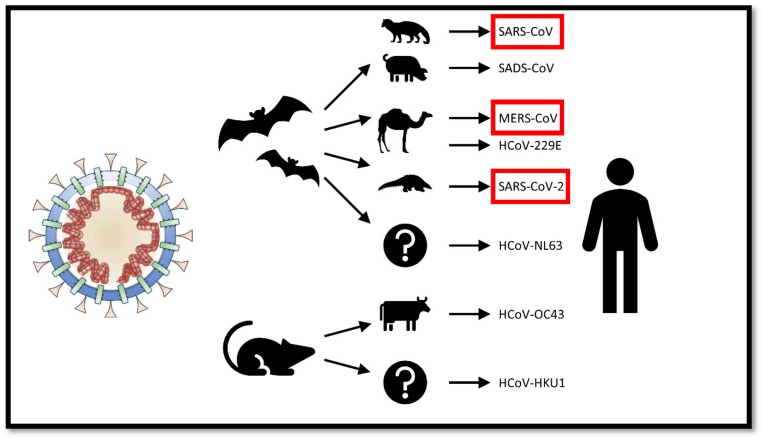
Animal origins of human coronaviruses. Severe acute respiratory syndrome coronavirus (SARS-CoV) and Middle East respiratory syndrome coronavirus (MERS-CoV) and were transmitted to humans from bats by civet cats and dromedary camels, respectively. The 2019 SARS-CoV-2 was likely transmitted to humans through pangolins that are illegally sold in Chinese markets [16,17].

**Figure 3 pathogens-09-00231-f003:**
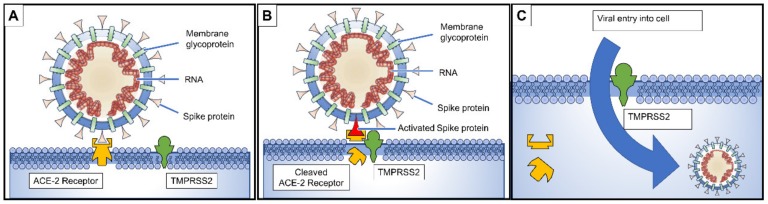
(**A**) Spike proteins on the surface of the coronavirus bind to angiotensin-converting enzyme 2 (ACE-2) receptors on the surface of the target cell; (**B**) The type II transmembrane serine protease (TMPRSS2) binds to and cleaves the ACE-2 receptor. In the process, the spike protein is activated; (**C**) Cleaved ACE-2 and activated spike protein facilitate viral entry. TMPRSS2 expression increases cellular uptake of the coronavirus [20,21,22].

**Figure 4 pathogens-09-00231-f004:**
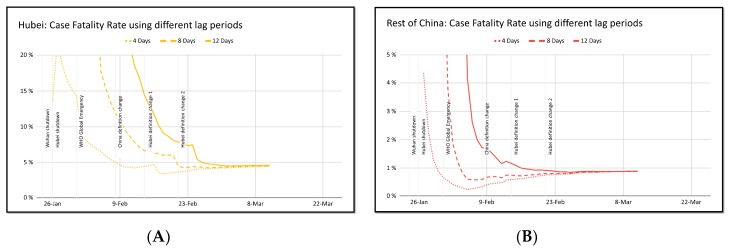
Estimating case fatality rate using different lag periods in (**A**) Hubei and (**B**) the rest of China. Credit to Nucleuswealth.com [14].

**Table 1 pathogens-09-00231-t001:** Epidemiologic characteristics of the first 44,672 confirmed cases in China [26].

	No. Cases (%)	Deaths (%)	CFR (%)
Overall	44,672	1023	2.3
Age			
0–9 yrs	416 (0.9)	-	-
10–19 yrs	549 (1.2)	1 (0.1)	0.2
20–29 yrs	3619 (8.1)	7 (0.7)	0.2
30–39 yrs	7600 (17.0)	18 (1.8)	0.2
40–49 yrs	8571 (19.2)	38 (3.7)	0.4
50–59 yrs	10,008 (22.4)	130 (12.7)	1.3
60–69 yrs	8583 (19.2)	309 (30.2)	3.6
70–79 yrs	3918 (8.8)	312 (30.5)	8.0
≥ 80 yrs	1408 (3.2)	208 (20.3)	14.8
Wuhan related exposure			
Yes	31,974 (85.8)	853 (92.8)	2.7
No	5295 (14.2)	66 (7.2)	1.2
Case Severity			
Mild	36,180 (80.9)	-	-
Severe	6168 (13.8)	-	-
Critical	2087 (4.7)	1023 (100)	49.0
Missing	257 (0.6)	-	-

**Table 2 pathogens-09-00231-t002:** Fatality rate by age, sex, and pre-existing medical conditions. The death rate represents the probability (%) of the corresponding group of dying from SARS-CoV-2 [26].

	CFR (%)
Age	
0–9 yrs	-
10–19 yrs	0.2
20–29 yrs	0.2
30–39 yrs	0.2
40–49 yrs	0.4
50–59 yrs	1.3
60–69 yrs	3.6
70–79 yrs	8.0
≥ 80 yrs	14.8
Sex	
Male	2.8
Female	1.7
Preexisting Medical Condition	
Cardiovascular disease	10.5
Diabetes	7.3
Chronic respiratory disease	6.3
Hypertension	6.0
Cancer	5.6
No preexisting condition	0.9

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
