# Peer review of "SARS-CoV-2 and Coronavirus Disease 2019: What We Know So Far"

_pathogens, 2020, doi:10.3390/pathogens9030231_

Round 1

Reviewer 1 Report

This study presents a summary of current knowledge regarding the novel coronavirus and the disease it causes. Readers of the scientific community should be interested in this topic. Several issues need to be addressed in this manuscript.

Several suggestions:

  1. Several references labeled as [Error! Reference source not found.] possibly by the Office should be corrected.
  2. Some of figures and tables were not mentioned in the text. They should be mentioned.
  3. Figures and Tables should be drawn by the authors and contain the references, which is lacking in the manuscript.
  4. Line 53, [full gene]. However, reference 5 only have the partial gene. Please check it.
  5. Line 33, after [order]. It is better to have a reference.
  6. Line 92. It is better to have a reference at the end of the sentence.
  7. Line 154. It is better to have a reference at the end of the sentence.
  8. Line 157. It is better to have a reference at the end of the sentence.
  9. Line 207. It is better to have a reference at the end of the sentence.
  10. Line 251. It is better to have a reference at the end of the sentence.
  11. Line 267. It is better to have a reference at the end of the sentence.

Author Response

Thank you for your suggestions, each of which was addressed as follows:

  1. Error references fixed.
  2. All figures and tables were reviewed and their references made clear. In particular, we referred to figure 3 separately as A, B, and C to help clarify. 
  3. Figures 1, 2, and 3 are original and drawn by the first author. Figure 4 is from an opensource online and proper reference is made in the text. We clarified the sourcing within the caption as well based on your recommendation. We also added references to the captions of all of the figures.
  4. Proper reference added
  5. Reference added
  6. Reference added. Sentence reworded for clarification.
  7. The sentence "...although MERS and SARS had higher mortality ..." was one we made based on inferences from previously cited data.
  8. We reworded the sentence to clarify that it is our judgement. No reference needed.
  9. Reference added
  10. Reference added
  11. Reference added

Thank you.

Reviewer 2 Report

This is a well written brief review of SARS-CoV2 virus epidemic based on most recent information available.

References to Tables and Figures in text need to be corrected.

Include Reference for Andersen et al #16 in line 92

Please be consistent with the date genome sequence was published  (9 Jan 2020) in Fig 1. It is mentioned correctly in Abstract.

Line  322 should be RBD not NBD

Author Response

Thank you for your review and your suggestions. We've addressed each of your suggestions as follows:

  1. References to tables and figures corrected. We apologize for the oversight.
  2. Proper reference to Andersen et. al added.
  3. Figure updated with correct genome sequence date (Jan 9, 2020)
  4. Line 322 corrected from RBD and NBD

Thank you.

Round 2

Reviewer 1 Report

This revised manuscript has addressed my previous suggestions properly.